# LVSPM: Long sequence view synthesis and pose estimation model

## Abstract

We present LVSPM, a generalizable model that jointly estimates camera poses and synthesizes novel views from uncalibrated image collections. Unlike prior approaches that rely on dense geometric supervision, LVSPM is trained only with RGB images and pose supervision, avoiding the need for dense 3D ground truth. LVSPM employs test-time training (TTT) layers, enabling efficient compression of tokens into fixed-size hidden states and scaling seamlessly to hundreds of input views. Experiments on RealEstate10k, Co3Dv2 and DL3DV, LVSPM surpasses VGGT in pose estimation across 10–256 input views. For novel view synthesis, LVSPM achieves state-of-the-art results in pose-free long-sequence rendering of the large baseline dataset DL3DV, and even exceeds pose-dependent models.

## 1 Introduction

Novel view synthesis (NVS) has been a long-standing challenge and a crucial driving force in 3D vision and graphics research. From per-scene optimization-based pipelines (Mildenhall et al., 2021; Müller et al., 2022; Chen et al., 2022; Barron et al., 2023; Kerbl et al., 2023; Yu et al., 2024; Huang et al., 2024) to feed-forward approaches (Yu et al., 2021; Chen et al., 2021; Charatan et al., 2024; Chen et al., 2025; Hong et al., 2024; Zhang et al., 2024; Jin et al., 2025), the community has developed diverse 3D representations and model architectures in recent years in pursuit of increasingly efficient, accurate, and scalable solutions. However, nearly all existing methods assume access to known camera poses, typically estimated through structure-from-motion (SfM) pipelines (Schonberger & Frahm, 2016; Snavely et al., 2006), which are computationally expensive and often fragile in real-world scenarios.

To avoid the reliance on known poses, several recent attempts have explored pose-free view synthesis with feed-forward models (Ye et al., 2024; Zhang et al., 2025a; Jiang et al., 2025b). While promising, these methods are mostly restricted to sparse-view input and cannot scale to long sequences, which are crucial for capturing wide scene coverage and enabling immersive experiences. In parallel, geometry-based models such as DUSt3R and VGGT (Wang et al., 2024; 2025a) have shown strong feed-forward performance on pose estimation, but they do not address view synthesis and rely on dense 3D supervision such as point maps that are far more expensive and less scalable to obtain than the image supervision used in classical view synthesis.

This leaves open the challenge of achieving long-sequence pose-free view synthesis without requiring heavy dense 3D labels. In this work, we address this challenge by introducing **a novel feed-forward framework, LVSPM, that jointly achieves long-sequence novel view synthesis and camera pose estimation from unposed inputs**, using only RGB image and camera pose supervision. As illustrated in Fig. 1, our approach produces high-fidelity novel renderings and accurate camera poses from hundreds of captured images of a large real scene, achieving practical and scalable long-sequence view synthesis with integrated camera calibration.

To this end, our design philosophy follows the spirit of LVSM (Jin et al., 2025), aiming to minimize 3D inductive biases by framing the task as sequence-to-sequence token prediction. Specifically, given a sequence of unposed input views, our model tokenizes the images and directly predicts tokens corresponding to both input-view camera parameters and novel-view RGB images, without relying on explicit 3D representations or handcrafted geometric modules. To scale this to long sequences, we adopt a LaCT backbone(Zhang et al., 2025b)–a recent test-time training (TTT) architecture that performs large-chunk TTT updates for efficient and scalable long-sequence modeling.

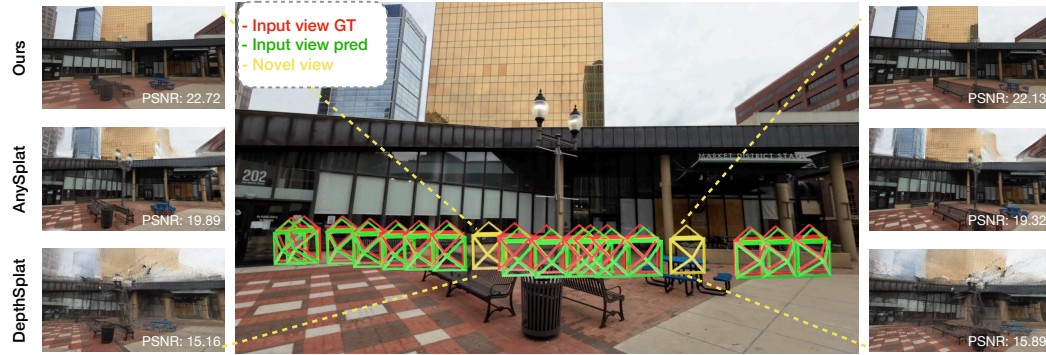

Figure 1: **LVSPM** can jointly achieve high-quality novel view synthesis and accurate camera pose estimation from uncalibrated long-sequence (128) multi-view images. A subset of the 128 input views with corresponding prediction and ground-truth poses is shown in the middle, while novel view synthesis comparisons are presented on the side. Our estimated poses closely align with ground truth, while our pose-free synthesis results surpass recent baselines, including the pose-free AnySplat (Jiang et al., 2025b) and even the pose-required DepthSplat (Xu et al., 2025b).

In addition, inspired by VGGT, we incorporate per-view camera tokens into the LaCT architecture, enabling direct pose estimation within the same sequence modeling framework. Together, these components lead to a novel feed-forward system that unifies long-sequence view synthesis and pose estimation within a minimal-bias, scalable framework.

We train our model on multiple large-scale synthetic and real datasets using only RGB images and camera pose supervision. Our model is trained with input sequences of up to 64 images and demonstrates effective scalability to as many as 256 inputs during inference, achieving photo-realistic rendering quality together with accurate pose estimation. Experimental results show that our approach leads to state-of-the-art performance on pose-free view synthesis across diverse and challenging benchmarks. It consistently outperforms existing pose-free methods in their relatively sparse-view settings and, more importantly, extends naturally to long-sequence, wide-coverage synthesis that prior pose-free baselines cannot handle. Notably, our pose-free rendering quality even matches or exceeds that of some pose-required feed-forward models like DepthSplat (Xu et al., 2025b). At the same time, our pose estimation results are also highly competitive: on multiple datasets, they match or surpass the performance of state-of-the-art geometry-based models such as VGGT, despite our method not relying on any additional dense 3D supervision. These findings highlight the feasibility of addressing fundamental 3D vision problems with cheaper supervision, opening new possibilities for scalable and generalizable 3D perception systems.

## 2 RELATED WORK

**Novel View Synthesis.** View synthesis has been extensively studied for decades in computer vision and graphics (Levoy & Hanrahan, 1996; Gortler et al., 1996; Debevec et al., 1996; Buehler et al., 2001; Zhou et al., 2018; Mildenhall et al., 2021; Kerbl et al., 2023; Hong et al., 2024; Sajjadi et al., 2022; Jin et al., 2025). In recent years, NeRF (Mildenhall et al., 2021), 3D Gaussian Splatting(3DGS) (Kerbl et al., 2023), and many variants of such 3D representations (Müller et al., 2022; Xu et al., 2022; Sun et al., 2022; Chen et al., 2022; Fridovich-Keil et al., 2022; Huang et al., 2024; Yu et al., 2024) with differentiable rendering have achieved photo-realistic results through end-to-end optimization, albeit at the cost of significant computational overhead per scene. To enable faster inference, numerous feed-forward methods have been developed for instant scene reconstruction and rendering, most of which rely on 3D representations and 3D-related architectural designs such as plane-sweep volumes (Chen et al., 2021; Johari et al., 2022; Zhang et al., 2022; Chen et al., 2025; Liu et al., 2024) or epipolar priors (Yu et al., 2021; Wang et al., 2021; Charatan et al., 2024; Suhail et al., 2022). Recently, large reconstruction models (LRMs) (Hong et al., 2024; Li et al., 2023; Wang et al., 2023; Zhang et al., 2024; Wei et al., 2024) have begun to reduce such architectural-level 3D biases, leveraging pure transformer architectures, while LVSM (Jin et al., 2025) further eliminates representation-level bias and achieves state-of-the-art view synthesis quality. Our method inherits

this minimal-bias philosophy but extends it to the joint problem of view synthesis and pose estimation, whereas most existing approaches still assume known camera poses.

On the other hand, most prior feed-forward methods remain restricted to sparse input views. Recent works such as Long-LRM (Ziwen et al., 2024) and LaCT (Zhang et al., 2025b) have begun to explore long-sequence inputs with tens or even hundreds of images using architectures like Mamba (Gu & Dao, 2023; Dao & Gu, 2024) and TTT (Sun et al., 2024); however, these approaches still assume known camera poses, which limits their practicality. Our approach builds upon LaCT and extends it to the pose-free setting, enabling joint long-sequence view synthesis and pose estimation.

**Camera Pose Estimation.** Camera pose estimation has traditionally relied on Structure-from-Motion (SfM) pipelines (Schonberger & Frahm, 2016; Snavely et al., 2006), which remain robust for large-scale reconstructions but suffer from heavy computational cost and failure modes in textureless regions or repetitive structures. These issues persist even with the advent of learning-based feature extractors (DeTone et al., 2018; Dusmanu et al., 2019; Revaud et al., 2019) and matchers (Sarlin et al., 2020; 2019; Liu et al., 2021). Recently, many learning-based approaches have attempted to directly regress camera poses (Lin et al., 2023; Rockwell et al., 2022; Cai et al., 2021), though the current state of the art comes from geometry-driven transformer-based models (Wang et al., 2024; 2025a). In particular, DUSt3R (Wang et al., 2024) formulates pairwise 3D reconstruction as point map regression, enabling pose-free 3D reconstruction and subsequent camera estimation via PnP. Numerous follow-up works have extended DUSt3R, improving its inference quality and scalability to longer sequences (Yang et al., 2025; Yuan et al., 2025; Tang et al., 2025; Wang et al., 2025c; Leroy et al., 2024). More recently, VGGT (Wang et al., 2025a) represents a major step forward, introducing a feed-forward multi-task transformer that jointly predicts cameras, depth, and point maps, achieving state-of-the-art performance across several geometry tasks, including pose estimation. However, these approaches primarily focus on geometric reconstruction, depend heavily on dense 3D labels, and do not directly address the problem of view synthesis. Our method takes inspiration from VGGT's camera estimation mechanism but integrates it into a view synthesis framework, achieving comparable or superior pose estimation results while eliminating the need for dense geometric ground truth.

**Pose-Free View Synthesis.** Many recent works have sought to remove the requirement of known camera poses in view synthesis to improve practicality. Early attempts integrated pose estimation into optimization-based frameworks such as NeRF, jointly estimating poses and scene representations during training (Lin et al., 2021; Bian et al., 2023; Chen et al., 2023; Truong et al., 2023; Xia et al., 2022). Recently, feed-forward approaches have gained traction. While some methods adopt a two-stage pipeline that first estimates cameras and then performs view synthesis (Jiang et al., 2022; 2025a; Zhang et al., 2025a), others aim to predict poses and novel views jointly. Early pose-free approaches typically achieve only sparse-view, object-level reconstruction and rendering (Wang et al., 2023; Jiang et al., 2023; Xu et al., 2024a). Building on the success of DUSt3R and 3D Gaussian Splatting, subsequent works have extended this idea to scene-level pose-free view synthesis by directly reconstructing Gaussian point clouds from unposed inputs and recovering poses via PnP (Ye et al., 2024; Smart et al., 2024; Xu et al., 2024b; Fan et al., 2024; Huang & Mikolajczyk, 2025). However, these approaches generally rely on dense geometric supervision and remain limited to only a handful of input views (often fewer than ten). More recently, AnySplat (Jiang et al., 2025b) extends VGGT to the view synthesis task, supporting several tens of input views, but it depends on VGGT pretraining and still requires dense 3D labels for supervision. In contrast, our method outperforms AnySplat under its setting and scales effectively to hundreds of input images. Notably, our model is trained from scratch using only RGB image and camera pose supervision, without requiring additional dense 3D geometric labels.

## 3 METHOD

Our work, LVSPM, introduces a feed-forward model designed to jointly estimate camera poses and synthesize novel views from a sequence of unposed images. The model architecture minimizes explicit 3D inductive biases, framing the problem as a sequence-to-sequence prediction task. We leverage a Large-Chunk Test-Time Training (LaCT) backbone (Zhang et al., 2025b) to efficiently process long input sequences, coupled with learnable camera tokens for direct camera pose and intrinsic parameter regression. This section details the model's input representation, architecture, and the supervision strategy used for training.

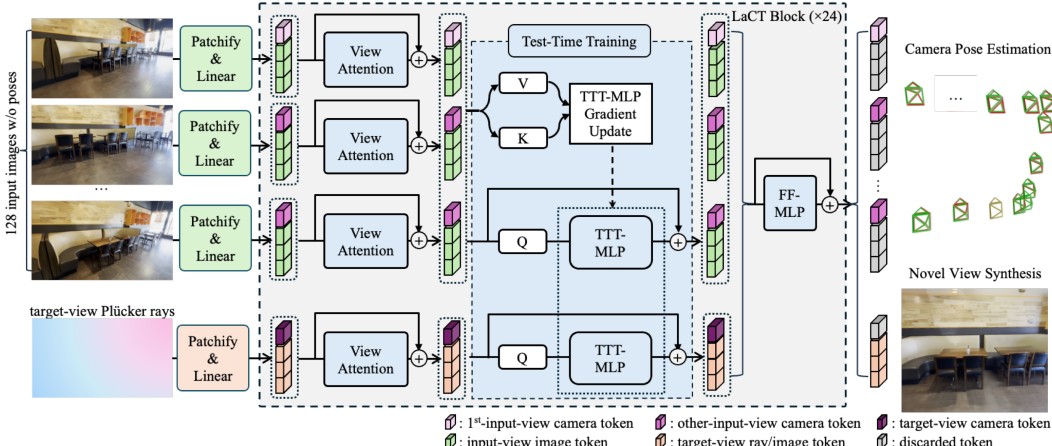

Figure 2: **Pipeline.** Our approach takes uncalibrated long-sequence input views, where each image is tokenized into patch tokens and augmented with a learnable camera token, while target views are represented by Plücker-ray tokens. All tokens are processed through a stack of LaCT blocks that combine per-view self-attention, an MLP-based TTT layer, and a feed-forward MLP layer. From the final tokens, our lightweight decoders produce novel-view RGB images and camera parameters, enabling accurate pose estimation together with high-quality long-sequence view synthesis.

## 3.1 PROBLEM FORMULATION AND OVERVIEW

Given a set of $N$ uncalibrated input images $\mathcal{I} = \{I_i\}_{i=1}^N$, where $I_i \in \mathbb{R}^{H \times W \times 3}$, our goal is to:

- Estimate camera poses $\{E_i = (t_i, r_i)\}_{i=1}^N$, as well as intrinsic parameters $\{K_i = (f_i^x, f_i^y)\}_{i=1}^N$, where $t_i \in \mathbb{R}^3$ is the 3-dim translation, $r_i \in \mathbb{R}^4$ is the 4-dim quaternion, and $\hat{f}_i^x, \hat{f}_i^x$ are scalars denoting the normalized camera focal lengths;

- Synthesize novel view $I^t$ from target novel camera configurations $(E^t, K^t)$.

This joint formulation eliminates the dependency on pre-computed structure-from-motion (SfM) pipelines while enabling end-to-end learning of geometric and appearance modeling.

As shown in Fig. 2. LVSPM consists of $L$ stacked LaCT blocks, each containing three main components: (1) a large-chunk test-time training (TTT) layer for long-range multi-view reasoning, (2) a window attention layer for local spatial dependencies, and (3) a feed-forward MLP network for channel mixing. The architecture processes variable-length sequences up to 1M tokens through an efficient test-time adaptation mechanism.

## 3.2 MODEL DETAILS

**Input representation and tokenization.** Unlike prior pose-required view synthesis models (Zhang et al., 2024; Jin et al., 2025) that typically condition both input and target views on Plücker rays (Plucker, 1865), our pose-free approach ignores input-view Plücker encoding and instead introduces additional learnable camera tokens to extract camera information. Specifically, each input image $I_i$ is divided into non-overlapping patches of $p \times p$ pixels, noted as $\{I_{ij} \in \mathbb{R}^{p \times p \times 3} | j = 1, \ldots, HW/p^2\}$. These patches are then directly projected to the model dimension $d$ with a linear layer: $x_{ij} = \text{Linear}_{input}(I_{ij})$. Inspired by VGGT (Wang et al., 2025a), we append each input image with a learnable camera token $c_i$ for pose prediction and intra-view context exchange. The first view is used as the canonical reference frame, assuming a zero translation and an identity rotation, and has a special initial learnable camera token $c_r$ to represent the reference frame. All other input views share another learnable camera token $c_x$. The translation scale is determined by normalizing the distance between the first and the furthest to unit length, providing a consistent reference frame across different scenes.

The input-view tokens $X_i$ for input view $i$ are represented as:

$$X_i = \{c_i, x_{i1}, x_{i2}, \ldots, x_{iM}\}, \ c_1 = c_r, \quad c_i = c_x \text{ for } i = 2, \ldots, N. \tag{1}$$

where $M = HW/p^2$. For simplicity, we denote by $X_{ij}$ an arbitrary token from input view $i$, where $X_{i0} = c_i$ corresponds to the camera token, and $X_{ij} = x_{ij}$ for $j > 0$ corresponds to the image tokens.

On the other hand, we still adopt Plücker rays to condition target views, providing the 3D camera context required for novel-view synthesis. More specifically, we compute the target frame Plücker rays from its camera parameters $(E^t, K^t)$. For the $k$-th target view, its Plücker ray patches are noted as $P_k^t = \{P_{kj}^t | j = 1, \ldots, HW/p^2\}$. These Plücker ray patches are transformed into tokens using another linear layer, similar to how we encode the image patches of the input views: $x_{kj}^t = \text{Linear}_{target}(P_{kj}^t)$. To be consistent with input view encoding, these target tokens are concatenated with another learnable token $c_t$ before feeding into the network. The target view tokens $T_k$ for target view $k$ is represented as:

$$T_k = \{c_k, x_{k1}^t, x_{k2}^t, \ldots, x_{kM}^t\}, \quad c_k = c_t \tag{2}$$

Similar to input-view token $X_{ij}$, we denote by $T_{kj}$ an arbitrary token from target view $k$.

**LaCT blocks.** We adopt the recent Large-Chunk Test-Time Training (LaCT) architecture (Zhang et al., 2025b) to process both input and target view tokens for joint view synthesis and pose estimation, effectively handling unordered, unposed multi-view inputs while remaining computationally efficient. In particular, our model consists of $L = 24$ LaCT blocks with alternating (windowed) self-attention, test-time training (TTT), and Feed-forward MLP layers; each TTT layer is equiped with a SwiGLU-MLP (Shazeer, 2020), noted as **TTT-MLP**$_l$, for test-time training.

Specifically, For the $l$-th block, the $HW/p^2 + 1$ tokens of each view are first fed into a per-view windowed self-attention layer:

$$\hat{X}_i^l = \text{Attn}_l(X_i^{l-1}) + X_i^{l-1}, \tag{3}$$

$$\hat{T}_k^l = \text{Attn}_l(T_k^{l-1}) + T_k^{l-1}. \tag{4}$$

Then we process multi-view tokens with the TTT layer, specifically:

$$q_{l,ij} = \text{Q}_l(\hat{X}_{ij}^l), \quad q_{l,kj}^t = \text{Q}_l(\hat{T}_{kj}^l), \tag{5}$$

$$k_{l,ij} = \text{K}_l(\hat{X}_{ij}^l), \tag{6}$$

$$v_{l,ij} = \text{V}_l(\hat{X}_{ij}^l), \tag{7}$$

where $Q_l$, $K_l$, and $V_l$ are learnable parameters to project the original tokens into Q, K, V parameters. The weight of TTT-MLP$_l$ is then updated with the gradient $G_l = \nabla_{\text{TTT-MLP}_l} ||v_l - \text{TTT-MLP}_l(k_l)||^2$ and a learnable learning weight $\eta$. The updated TTT-MLP$_l^{\text{updated}}$ is used to process the final outputs of this TTT layer for token update:

$$\tilde{X}_{ij}^l = \text{TTT-MLP}_l^{\text{updated}}(q_{l,ij}) + \hat{X}_{ij}^l, \tag{8}$$

$$\tilde{T}_{kj}^l = \text{TTT-MLP}_l^{\text{updated}}(q_{l,kj}^t) + \hat{T}_{kj}^l \tag{9}$$

Note that only input view tokens are sent through $\text{K}_l, \text{V}_l$ and generate gradient for TTT MLP updates, as done in Zhang et al. (2025b). This way, the target view tokens don't need to interact with one another, enabling efficient synthesis of each novel view independently. The token outputs from the $l$-th LaCT block come from a final feed-forward MLP, denoted as FF-MLP$_l$:

$$X_{ij}^l = \text{FF-MLP}_l(\tilde{X}_{ij}^l) + \tilde{X}_{ij}^l, \tag{10}$$

$$T_{kj}^l = \text{FF-MLP}_l(\tilde{T}_{kj}^l) + \tilde{T}_{kj}^l \tag{11}$$

**Prediction Heads.** After the 24 LaCT blocks, target view tokens are decoded with a two-layer MLP, denoted as $\text{MLP}_{\text{rgb}}$, and rearranged to form the final novel-view RGB predictions. On the other hand, in contrast to the heavy camera head used by VGGT (Wang et al., 2025a), we adopt simple light weight MLPs for camera paerameter decoding. Specifically, the camera pose of each view is decoded from the final camera token with a simple two-layer $\text{MLP}_{\text{pose}}$, where the output is 9-dim: a 4-dim quaternion, a 3-dim translation, and 2-dim $\hat{f}^x$ and $\hat{f}^y$ focal length concatenated. We show that such lightweight camera heads are already sufficient to produce accurate camera parameters, even surpassing VGGT in estimation accuracy.

Table 1: Pose estimation results (AUC ↑) across datasets and number of input views. The **best** is marked in bold, and the second best is marked with underline. Our method consistently outperforms baselines with particularly strong gains on long sequences.

| Datasets | # Views Method | 10 views AUC30 | AUC5 | AUC3 | 64 views AUC30 | AUC5 | AUC3 | 128 views AUC30 | AUC5 | AUC3 |
|---|---|---|---|---|---|---|---|---|---|---|
| Re10k | Fast3R | 72.08 | 29.42 | 18.06 | 71.02 | 27.61 | 16.65 | 69.22 | 25.02 | 14.57 |
| | Cut3R | 81.71 | 43.99 | 31.13 | 78.35 | 37.53 | 24.82 | 75.78 | 33.37 | 21.00 |
| | VGGT | 80.37 | 34.36 | 20.94 | 79.47 | 32.62 | 19.55 | 80.70 | 34.50 | 21.12 |
| | Ours | **91.51** | **64.62** | **51.42** | **92.42** | **66.25** | **53.69** | **92.53** | **66.37** | **53.65** |
| Co3dv2 | Fast3R | 77.26 | 36.94 | 23.52 | 80.20 | 40.00 | 25.70 | 77.90 | 36.30 | 21.60 |
| | Cut3R | 79.40 | 33.80 | 35.60 | 82.60 | 36.80 | 37.90 | 83.10 | 35.40 | 38.20 |
| | VGGT | **89.39** | **62.40** | **50.71** | 90.21 | 68.84 | 59.21 | 90.10 | 68.34 | 58.21 |
| | Ours | 88.09 | 60.10 | 47.11 | **91.04** | **70.73** | **61.08** | **91.31** | **71.86** | **62.71** |

Note that VGGT is a transformer-based architecture; to achieve camera estimation, VGGT introduces additional frame-attention modules coupled with full attention to process the camera token jointly with per-view image tokens. In contrast, our design leverages LaCT blocks, which natively incorporate local windowed attention per view, eliminating the need for extra modules. As a result, our framework achieves better camera estimation while using fewer model parameters.

### 3.3 TRAINING OBJECTIVE

The model is trained end-to-end with photometric, camera pose, and intrinsic losses:

$$\mathcal{L}_{\text{rgb}} = ||I_{\text{pred}} - I_{\text{gt}}||_2^2 + \lambda_{\text{LPIPS}} \cdot \text{LPIPS}(I_{\text{pred}}, I_{\text{gt}}), \tag{12}$$

$$\mathcal{L}_{\text{pose}} = ||t_{\text{pred}} - t_{\text{gt}}||_2^2 + \cdot ||r_{\text{pred}} - r_{\text{gt}}||_2^2, \tag{13}$$

$$\mathcal{L}_{\text{int}} = ||\hat{f}_{\text{pred}}^x - \hat{f}_{\text{gt}}^x||_2^2 + ||\hat{f}_{\text{pred}}^y - \hat{f}_{\text{gt}}^y||_2^2, \tag{14}$$

where $I_{\text{pred}}$, $t_{\text{pred}}$, $r_{\text{pred}}$, $\hat{f}_{\text{pred}}^x$, and $\hat{f}_{\text{pred}}^y$ are RGB image, camera translation, rotation quaternion, and x, y focal length predictions respectively, and the subscript $_{\text{gt}}$ denotes the ground-truth values for the corresponding quantities.

The total training loss is:

$$\mathcal{L}_{\text{total}} = \lambda_{\text{rgb}}\mathcal{L}_{\text{rgb}} + \lambda_{\text{pose}}\mathcal{L}_{\text{pose}} + \lambda_{\text{int}}\mathcal{L}_{\text{int}}.$$

where $\mathcal{L}$rgb, $\lambda$pose, and $\lambda_{\text{int}}$ are the weights for the RGB, pose, and intrinsic losses, respectively.

## 4 EXPERIMENTS

We evaluate LVSPM on multiple challenging benchmarks to demonstrate its effectiveness at joint camera pose estimation and view synthesis from unposed input views. For pose estimation, We evaluate on RealEstate10k (RE10k) (Zhou et al., 2018), Co3Dv2 (Reizenstein et al., 2021), and DL3DV-10K (Ling et al., 2024). For novel view synthesis, we evaluate on DL3DV-10K (Ling et al., 2024) and Tanks and Template Knapitsch et al. (2017b), comparing against both pose-free and pose-dependent baselines. We evaluate our method across varying numbers of input views- from sparse (10 views) to large-scale real-world sequences with 256 views.

### 4.1 EXPERIMENT SETTINGS

**Model Details.** Our model uses 24 layers of alternating view-attention and test-time-training layers. Model dimension $d = 768$. We use 12 heads for self-attention layers. We follow (Zhang et al., 2025b) to apply RoPE embedding (Su et al., 2024) on Q and K inputs.

**Training Details.** We implement LVSPM with PyTorch and train on 64 NVIDIA H100 GPUs. We train our model on a mix of large-scale synthetic and real-world datasets, including Aria Synthetic Environments (ASE) (Pan et al., 2023), DL3DV-10K (Ling et al., 2024), ScanNet++ (Yeshwanth et al., 2023), Hypersim (Roberts et al., 2021), and CO3Dv2 (Reizenstein et al., 2021). Our model is not trained on the evaluation split of these datasets. The model undergoes a three-stage training process. First, we pre-train on the synthetic ASE dataset for 60k iterations with 32 input and target

Table 2: Pose Estmation results on DL3DV-10K across 10–256 input views.

| DL3DV | 10 views | | | 128 views | | | 256 views | | |
|---|---|---|---|---|---|---|---|---|---|
| | AUC 30 | AUC5 | AUC 3 | AUC 30 | AUC5 | AUC 3 | AUC 30 | AUC5 | AUC 3 |
| Fast3r | 63.5 | 25.3 | 15.6 | 59.9 | 22.1 | 13.1 | 52.6 | 15.5 | 8.2 |
| Cut3r | 82.9 | 57.4 | 45.7 | 79.5 | 44.0 | 29.7 | 72.8 | 32.5 | 20.2 |
| VGGT | **94.5** | **88.2** | **84.1** | 95.39 | 88.7 | 84.6 | 95.2 | 87.9 | 83.5 |
| Ours | 93.7 | 87.8 | 82.9 | 95.17 | **89.1** | **85.95** | 95.2 | **89.33** | **86.17** |

views at $128 \times 128$ resolution. Second, we mix ASE with other training datasets (DL3DV-10K, ScanNet++, Hypersim, Co3Dv2), and train for another 60k iterations. Finally, we progressively increase input resolution to $512 \times 448$, and increase scale to 64 input and target views. More details can be found in the Appendix.

**Baselines.** We compare against recent pose estimation methods, namely Fast3R (Yang et al., 2025), Cut3R (Wang et al., 2025b), and VGGT (Wang et al., 2025a). For view synthesis, we compare with both pose-free methods AnySplat (Jiang et al., 2025b). Noposplat Ye et al. (2024) and pose-dependent method DepthSplat (Xu et al., 2025a). For fair comparison, we use official implementations where available and use the same evaluation datasets for all models.

**Evaluation Metrics.** For pose estimation, we follow VGGT (Wang et al., 2025a) to report Area Under Curve (AUC) metrics at 3-deg (AUC3), 5-deg (AUC5), and 30-deg (AUC30) thresholds. The pose AUC evaluates both rotation and translation accuracy jointly- a pose is considered correct at a given threshold only if both relative rotation error (RRE) and relative translation error (RTE) are under the corresponding threshold. Higher AUC values indicate more accurate pose estimation, with AUC3 being the strictest metric, requiring sub-3-degree precision for both rotation and translation. For view synthesis, we measure PSNR, SSIM (Wang et al., 2004), and LPIPS (Zhang et al., 2018) on held-out test views, reporting average metrics across scenes.

## 4.2 CAMERA POSE ESTIMATION

Our pose estimation results in Tab. 1 demonstrate LVSPM's strong performance across varying input view counts and datasets. On RealEstate10k (RE10k) with 10 views, we achieve 91.51 AUC30 compared to VGGT's 80.37, and significantly outperform Fast3R and Cut3R. The gap widens further with more views- at 64 or 128 inputs, we maintain our lead while VGGT's performance does not improve with more input views.

On Co3Dv2, which contains more diverse object categories, LVSPM achieves 88.09 AUC30 at 10 views, matching VGGT's 88.39. On Co3Dv2's AUC5 and AUC3 metrics, VGGT shows marginally better performance at 10 views, likely due to its dense 3D supervision providing stronger geometric constraints for fine-grained accuracy. However, our approach scales better with additional views. For 64 or 128 views, our model outperforms all baselines including VGGT, demonstrating superior scalability to longer sequences.

Most impressively, on the challenging DL3DV dataset, which features large-scale indoor and outdoor scenes with large baselines, LVSPM demonstrates exceptional scalability. With 10 views, we match VGGT. At 128 views, our method continues to improve while VGGT shows slight degradation. These results validate that joint training with view synthesis provides strong supervisory signals for camera estimation, even without dense 3D supervision.

## 4.3 VIEW SYNTHESIS

Table 3 shows that LVSPM establishes new state-of-the-art results for pose-free long-sequence view synthesis on DL3DV-10K. With only 16 input views, which is very sparse considering the scene scale, we achieve 18.91 PSNR, outperforming AnySplat by significant margins, while also achieving slightly better performance than DepthSplat, which requires explicit pose inputs. Our method demonstrates exceptional scaling- at 32 views, we reach 20.25 PSNR compared to DepthSplat's 17.81 and AnySplat's 17.70.

The performance gap becomes even more pronounced with longer sequences, while other baselines show little improvement or even degradation on view synthesis quality. At 64 views, LVSPM achieves 21.40 PSNR, far exceeding AnySplat's 18.81 PSNR, while DepthSplat sees a performance drop. Most remarkably, with 128 input views where DepthSplat cannot operate, we achieve 22.16 PSNR, a more than 3dB gain over AnySplat. The consistent improvement with more views (PNSR

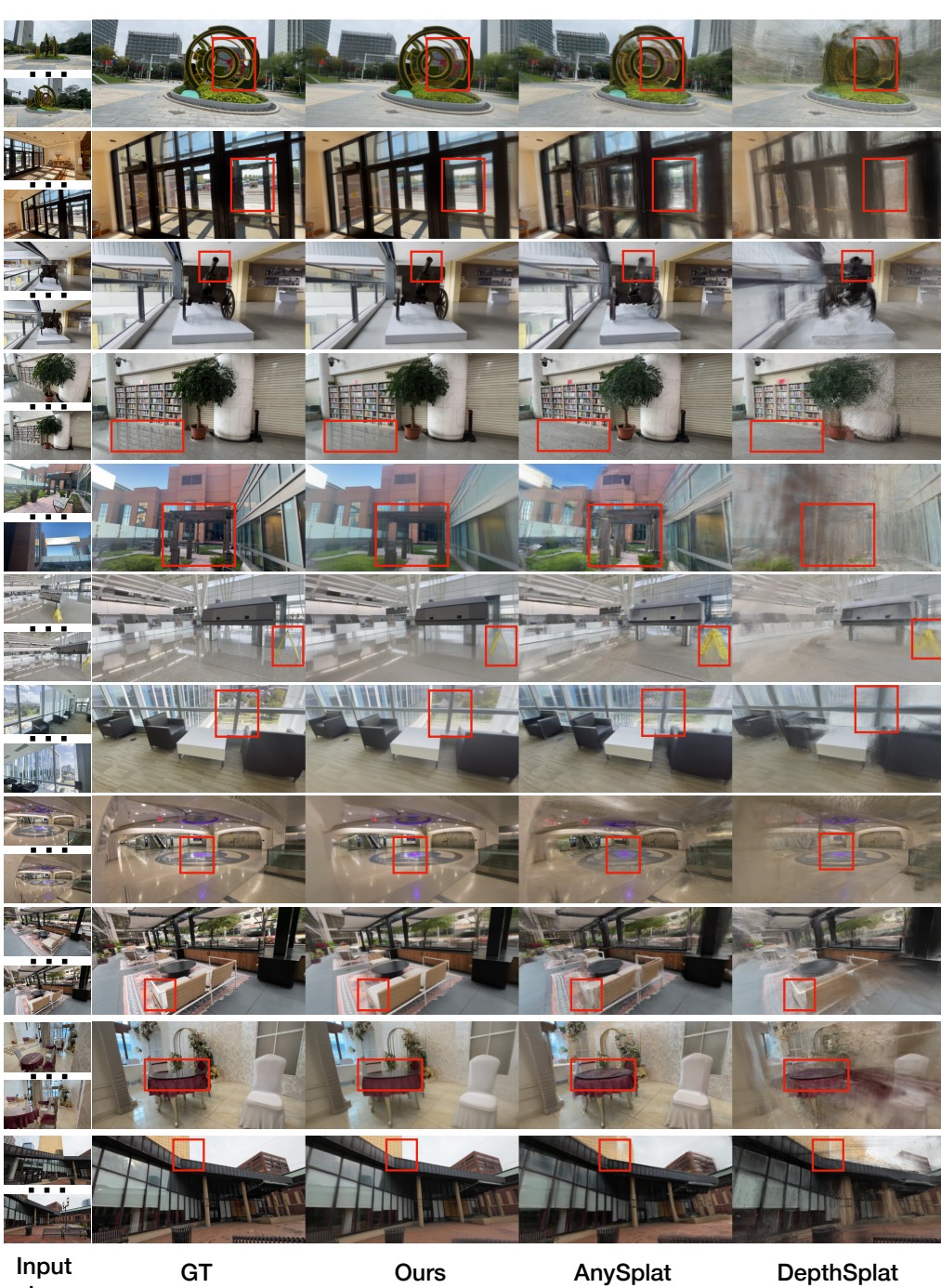

Figure 3: **Novel-view rendering comparison on the DL3DV dataset with long-sequence input.** We visualize results for 128 inputs using our method and AnySplat Jiang et al. (2025b), while DepthSplat Xu et al. (2025b) uses its maximum of 64. Unlike DepthSplat Xu et al. (2025b), which requires known poses, our method and AnySplatt Jiang et al. (2025b) are pose-free.

$18.91 \rightarrow 20.25 \rightarrow 21.40 \rightarrow 22.16$) demonstrates effective utilization of long sequences, while competing methods show much smaller gains or even degradation.

Our LPIPS scores show particularly strong perceptual quality, improving from 0.319 at 16 views to 0.215 at 128 views, indicating that our method produces increasingly realistic renderings as more input views become available. This aligns with the qualitative results shown in Fig. 3, our method ren-

Table 3: Novel view synthesis on DL3DV-10K (PSNR↑ / SSIM↑ / LPIPS↓) across 16–128 input views. The **best** is mark in bold. DepthSplat requires known poses; AnySplat and ours are pose-free.

| | 16 Views | | | 32 Views | | | 64 Views | | | 128 Views | | |
|---|---|---|---|---|---|---|---|---|---|---|---|---|
| | PSNR | SSIM | LPIPS | PSNR | SSIM | LPIPS | PSNR | SSIM | LPIPS | PSNR | SSIM | LPIPS |
| DepthSplat | 18.43 | **0.612** | 0.334 | 17.81 | **0.596** | 0.356 | 16.58 | 0.541 | 0.421 | | - | |
| AnySplat | 16.13 | 0.440 | **0.395** | 17.70 | 0.499 | 0.352 | 18.81, | 0.555 | 0.321 | 19.14 | 0.574 | 0.314 |
| Ours | **18.91** | 0.521 | 0.319 | **20.25** | 0.590 | **0.265** | **21.40** | **0.645** | **0.231** | **22.16** | **0.669** | **0.215** |

Table 4: Novel view synthesis on Tanks and template. Our methods outperform the baseline on both small and large camera movements, as well as sparse and dense views.

| | 6 Views (Small) | | | 12 Views (Small) | | | 64 Views (Large) | | |
|---|---|---|---|---|---|---|---|---|---|
| | PSNR | SSIM | LPIPS | PSNR | SSIM | LPIPS | PSNR | SSIM | LPIPS |
| Nopoplat | 15.22 | 0.359 | 0.529 | 14.31 | 0.377 | 0.563 | 12.49 | 0.328 | 0.693 |
| AnySplat | 16.44 | **0.471** | 0.327 | 19.45 | 0.634 | 0.226 | 17.69 | 0.512 | 0.348 |
| Ours | **17.60** | 0.460 | **0.325** | **19.89** | 0.582 | 0.205 | **19.01** | 0.527 | 0.344 |

ders a sharper reflection, and does not suffer from layered surfaces as Anysplat Jiang et al. (2025b). The consistent improvement on LPIPS and visual quality indicates our method learns robust multi-view representations that generalize well to novel viewpoints, effectively leveraging the additional context from longer sequences.

We also evaluate our methods on the challenging Tanks and Template(TnT) Knapitsch et al. (2017a) dataset. We evaluate both small camera movement and large camera movement. As shown in Fig. 4, our methods achieve better results on 6-64 inputs with large or small camera movements.

## 4.4 ABLATION STUDY

We conduct experiments on the DL3DV dataset to validate our design choices. Due to limited resources, we conduct experiments on image resolution of 128 X 128. We first validate the effect of NVS supervision on pose estimation "W/O RGB". As shown in

Table 5: Ablation Studies.

| Ablations | Pose | | | NVS | | |
|---|---|---|---|---|---|---|
| | AUC 30 | AUC5 | AUC 3 | PSNR | SSIM | LPIPS |
| W/o RGB | 83.3 | 51.5 | 38.2 | | - | |
| W/O Syn | 92.7 | 84.3 | 79.9 | 20.08 | 0.589 | 0.267 |
| Ours | **94.5** | **85.6** | **80.2** | **21.12** | **0.638** | **0.222** |

Table 5, pose accuracy degrades without novel view supervision. Indicating our choice of NVS supervision is crucial for accurate pose estimation, alleviating the need for dense 3D supervision. We also experimented on the importance of synthetic data, where we removed synthetic data pertaining and directly trained on real data, denoting as "W/O Syn". Results in Table 5 show notable improvement on both pose estimation and novel view rendering.

## 5 CONCLUSION AND LIMITATIONS

We presented LVSPM, a unified framework that tackles the long-standing challenge of pose-free view synthesis at scale. Our approach demonstrates that joint camera pose estimation and novel view synthesis can be effectively learned from RGB images, eliminating the need for expensive dense 3D supervision while matching or exceeding methods that rely on it. LVSPM is the among the first pose-free view synthesis model that scales effectively to long sequences of 200+ views, enabled by the efficient test-time training mechanism, and produces particularly strong results on challenging large-scale benchmarks such as DL3DV-10K. LVSPM represents a significant step toward practical, scalable 3D scene understanding from unposed image collections, bringing us closer to systems that can operate on real-world data without expensive preprocessing or annotation requirements.

Despite these advances, limitations remain. The multi-stage training process adds complexity. Performance can degrade on scenes with extreme lighting changes or minimal texture, where even implicit geometric reasoning becomes challenging. Additionally, while our pose estimation is competitive, specialized geometry-focused methods may still excel in certain edge cases. Our future directions include exploring full self-supervised training to further reduce supervision requirements in the training dataset. The success of our minimal-supervision approach also opens questions about the role of explicit 3D representations in modern vision systems. Our method and results open a new avenue for future 3D research to rely on less structured priors, so that the training dataset and model size could be future scaled.

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

## A  APPENDIX

### A.1  LLM USAGE

In line with ICLR policy, we used an LLM strictly as a writing assistant, and to suggest candidate related-work papers; all text, claims, and citations were authored, verified, and curated by the authors. No confidential submission content was provided to an LLM. The research concepts, experimental design, analysis, and conclusions were entirely developed by the authors without substantive contribution from LLMs.

### A.2  ETHICS STATEMENT

This work focuses on developing a feed-forward model for long-sequence pose-free view synthesis and camera pose estimation using publicly available datasets (e.g., RealEstate10k, Co3Dv2, DL3DV-10K, ASE, ScanNet++, Hypersim). All datasets used are standard computer vision benchmarks with appropriate licenses and have been widely employed in prior literature. Our research does not involve human subjects, personally identifiable information, or sensitive content. No proprietary or restricted data were collected, and no personally identifiable or biometric information was used. The project adheres to the ICLR Code of Ethics: we respect privacy and security, ensure reproducibility, and avoid discriminatory or harmful applications. The methods developed here are intended for scientific and educational purposes only; any downstream applications (e.g., photorealistic scene rendering) should be deployed responsibly to avoid misuse such as privacy invasion or malicious surveillance.

### A.3  REPRODUCIBILITY STATEMENT

We have taken multiple steps to ensure reproducibility of our results. All architectural details of the LVSPM model, including the LaCT backbone, tokenization scheme, loss functions, and training schedules, are fully described in the Method and Experiments sections. Comprehensive training settings (datasets, resolutions, optimizer, learning rate, and number of iterations) are provided in Section 4.1. Evaluation metrics and protocols for camera pose estimation (AUC3/5/30) and view synthesis (PSNR/SSIM/LPIPS) are specified in the experimental setup. Source code and pretrained models will be released upon acceptance to enable independent verification of all reported numbers. Together, these details allow researchers to reproduce our training procedure, replicate our benchmarks, and extend our work to new datasets.

### A.4  MORE IMPLEMETATION DETAILS

We use AdamW optimizer (Loshchilov & Hutter, 2019) with an initial learning rate of 1e-4 and weight decay of 0.05. The batch size of the first and second stages is 12 batches per GPU. During the resolution up-scaling, we always keep nearly 500K tokens on each GPU and decrease the batch size when the resolution is higher. The loss weights are set to $\lambda_{\mathrm{rgb}} = 1.0, \lambda_{\mathrm{pose}} = 0.5, \lambda_{\mathrm{int}} = 0.5, \lambda_{\mathrm{LPIPS}} = 0.5$.

### A.5  QULITATIVE NVS RESULTS ON THE TNT DATASET

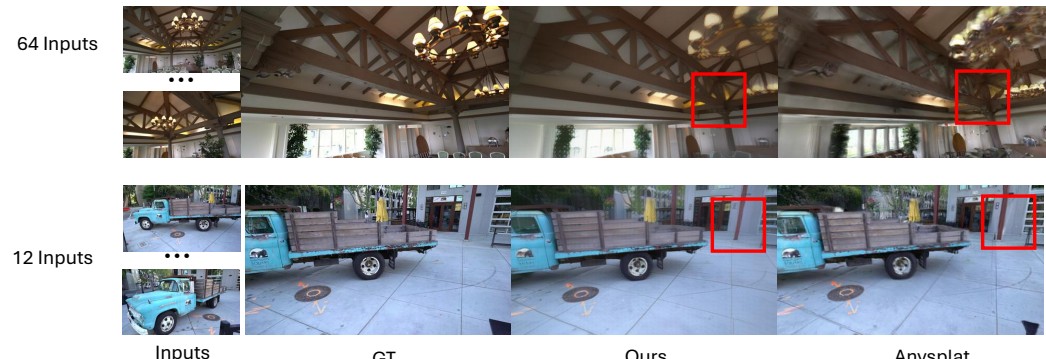

Figure 4: **Novel view rendering comparison on the TnT Knapitsch et al. (2017a) dataset.**

