# OpenReview forum: "LVSPM: Long Sequence View Synthesis and Pose Estimation Model"
_ICLR.cc/2026/Conference — Submitted to ICLR 2026_

### Official Review · Reviewer_BSiH · 2025-10-30

**Soundness:** 3
**Presentation:** 2
**Contribution:** 3
**Rating:** 4
**Confidence:** 4

**Summary:**

The paper presents LVSPM, a transformer-based model for long-sequence pose-free novel view synthesis (NVS) and camera pose estimation.
Unlike prior approaches that require dense 3D supervision (e.g., depth or point maps), LVSPM only uses RGB and pose supervision.
It extends the LaCT (Large-Chunk Test-Time Training) architecture to handle hundreds of unposed input views efficiently.
The model achieves state-of-the-art performance on several datasets such as DL3DV-10K, RealEstate10k, and Co3Dv2, outperforming VGGT in pose estimation and AnySplat in NVS.

**Strengths:**

A unified framework that jointly performs pose estimation and pose-free NVS at scale.

Can handle up to 256 unposed input images efficiently via the LaCT backbone.

Outperforms VGGT in pose estimation and AnySplat/DepthSplat in view synthesis across multiple datasets.

**Weaknesses:**

- Limited Generalization Evaluation


The experiments are mostly conducted on in-domain datasets, especially DL3DV-10K.

At this scale of training (64×H100 GPUs), more zero-shot evaluations are expected to demonstrate generalization.


- Sparse-View Unexplored


The paper evaluates primarily on semi-dense inputs (6–128 views).

It remains unclear how LVSPM performs under very sparse conditions (1–6 views), which are critical for real-world deployment and fairness against other baselines, e.g. the sparse views are supported in VGGT.


- Inference Efficiency Analysis Missing

Since LaCT is the core contribution, the paper should include inference runtime scaling with the number of views (e.g., 16→256).



- RGB v.s. Geometry Scaling

Without need of geometry labels, e.g. depth, the propsoed method could scale more than baselines VGGT, how much benefit will gain from this? How's the scaling curve of the proposed method w.r.t the data scale.

**Questions:**

Please see the weakness section.

---

> ### Author Response · Authors · 2025-11-25
> **Author Response (part 1/1)**
>
> # R4 BSiH
>
> **Limited Generalization Evaluation**
>
> We add experiments on more out-of-domain datasets, including Re10K, MipNeRF 360, and LLFF. We kindly refer the reviewers to Common Question 2 and Common Question 3 for a comprehensive evaluation.
>
> **Sparse-View Evaluation**
>
> To test performance on sparse view settings, we conduct experiments on RE10k with 2 input views, we report results both zero-shot generalize and finetuned on RE10k; the results are in common question 2. Our model zero-shot generalize on RE10k is better than AnySplat and achieves the best results after finetuning.
>
> We also conduct zero-shot generalize to MipNeRF 360 and the LLFF dataset under 3 and 6 input views, results are reported in common question 3. Our model achieves better results than the previous pose-free method NoPoSplat and Flare, and outperforms MVSplat, which requires known pose input.
>
> **Inference runtime scaling with the number of views**
>
> We report inference time across different numbers of input views at a resolution of 512×288, we refer the reviewers to the common question 4 for detailed results. LVSPM scales linearly with the number of views and maintains a rendering speed of **58.8 FPS** regardless of input count.
>
>
> **RGB v.s. Geometry Scaling**
>
> The main advantage of removing the need for geometry labels (e.g., depth, point clouds, or mesh supervision) is that it significantly lowers the data-collection barrier. Obtaining camera poses is far cheaper and more scalable than generating dense 3D geometry for large multiview/video datasets. This makes our framework inherently more suitable for future large-scale data collection and training, especially in scenarios where high-quality geometric annotations at internet scale are impractical.
>
> In this paper, our focus is not to demonstrate massive data scaling, but rather to justify the effectiveness of the proposed design of avoiding geometry supervision. We show that under comparable (if not smaller) training data scales, our method achieves competitive or superior performance to VGGT, which relies on dense geometry labels.

---

### Official Review · Reviewer_83qC · 2025-10-30

**Soundness:** 3
**Presentation:** 2
**Contribution:** 2
**Rating:** 4
**Confidence:** 3

**Summary:**

This work extends Test-Time Training layers to do novel view synthesis without known input camera parameters. The plucker map of the conditional input views is removed. Instead, like VGGT, a camera token is attached to finally predict the camera pose of the corresponding views. To model is trained to map a given plucker map to their corresponding novel-view views. The ground truth camera poses are normalized in VGGT manner so the target-view poses can be determined for training. The effectiveness is verified on pose estimation and novel-view synthesis tasks.

**Strengths:**

The proposed method is a straightforward extension to the recent strong TTT models for unposed scenarios. Every module makes sense. The improvement is good comparing to the other recent works under the same unposed setup.

**Weaknesses:**

Technical contribution is limited. It's a simple modification to the original TTT by removing input pose condition and attaching VGGT-like camera token to predicting camera parameters. The new knowledge it brings is limited.

The reason why a synthetic dataset pertaining is needed unclear to me. The ground-truth for training is poses and images where the poses for real-world data are quite accurate already. However, the ablation says synthetic pertaining is needed. Discussion and analysis for this is needed.

**Questions:**

Minor comment. Some of the subscript/superscript in the equations can be skip for brevity. The current equations is somewhat hard to read.

---

> ### Author Response · Authors · 2025-11-25
> **Author Response (part 1/1)**
>
> **Technical Contribution**
>
> We appreciate the reviewer’s assessment that our modules make sense and acknowledgment of our performance improvement. In addition to the discussion on novelty and contribution in Common Question 1, we would like to further clarify the following regarding the characterization of our technical contribution as a "simple modification":
>
> 1. **Simplicity as a Strength:** We argue that replacing complex, hand-crafted geometric modules (like the dense correlation volumes or point-map heads in VGGT/DUSt3R) with a generic, unified token stream is a valuable technical insight. It aligns with the broader trend in computer vision (e.g., ViT vs. CNN) where removing inductive bias in favor of scalable sequence modeling yields better generalizability.
>
> 2. **The "Free Lunch" of NVS:** Our work provides the technical insight that the Novel View Synthesis acts as a powerful regularizer for pose estimation. As shown in Ablation Table 5, the RGB loss significantly boosts pose accuracy. This interaction between NVS and Pose in a feed-forward transformer is a novel technical finding that goes beyond simple architectural assembly.
>
>
> **Why use synthetic data pre-training (Real data pose is accurate already)**
>
> Real-world datasets (e.g., RealEstate10K) often exhibit limited camera baselines and uneven camera-pose distributions, largely reflecting common videography patterns such as forward-facing motion. Moreover, the pose accuracy of real-world data is constrained by COLMAP, which can suffer from drift, scale inconsistency, or failure in textureless or reflective regions. These factors collectively restrict the diversity and reliability of the supervision signal.
> In contrast, synthetic datasets (e.g., ASE) provide (1) perfectly accurate camera poses without reconstruction noise, and (2) much more diverse and complex trajectories spanning wide baselines and varied motion patterns. Such diversity helps stabilize early training of our “camera token” mechanism and prevents overfitting to real-world camera biases.
> As demonstrated in our ablations, synthetic pre-training improves both pose estimation and novel view synthesis performance. We will add this clarification to the revision.
>
> **Equations are hard to read**
> Thanks for your suggestion, we will revise the paper for better readability.

---

> > ### Comment · Reviewer_83qC · 2025-11-28
> >
> > Really appreciate the author response.
> >
> > I agree with author comments and also like simple and good performing methods. Regarding the free lunch of NVS, I don't think it is a new finding that rendering loss can help pose estimation. For instance, COLMAP-Free 3DGS, InstantSplat, or the most related NopoSplat use photometric loss to achieve better poses.
> >
> > I like new the discussion on why synthetic data pre-training is needed in this work, which addresses my main question. I will raise my rating to 6.

---

### Official Review · Reviewer_5sus · 2025-10-31

**Soundness:** 1
**Presentation:** 3
**Contribution:** 2
**Rating:** 2
**Confidence:** 5

**Summary:**

This paper presents LVSPM, a feedforward model that simultaneously performs novel view synthesis and camera pose estimation. It extends the implicit novel view synthesis models LVSM and LaCT by removing the pose embedding from the input and adding an additional pose estimation head to predict camera poses. LVSPM supports a variable number of input views. Experimental results demonstrate that the proposed method achieves state-of-the-art performance in both pose estimation and novel view synthesis.

**Strengths:**

- The proposed method achieves SOTA performance in both pose estimation and novel view synthesis.

- The paper is well written, clear, and easy to follow.

**Weaknesses:**

- The model uses an input resolution of 512 × 448, which differs from that used in previous pose estimation and novel view synthesis models such as VGGT, AnySplat, DepthSplat, and NoPoSplat. How does the proposed method compare against these methods?

- The paper lacks details about the evaluation setup. For novel view synthesis on DL3DV and Tanks and Temples, how are the input views sampled?

- The novelty of the paper is limited. It mainly combines the LaCT version of LVSM with VGGT. Since LaCT has already shown strong performance in novel view synthesis and VGGT in pose estimation, the paper provides little new insight.

- The claim that the proposed method outperforms DepthSplat is not well supported. DepthSplat was trained using a fixed 6-view input, which explains why the proposed method performs better when more views are provided. A comparison with DepthSplat under the same 6-view setting is missing.

- The paper does not evaluate novel view synthesis on the RealEstate10K dataset, which is widely used in this field.

**Questions:**

- How does the pose estimation performance compare with recent SOTA methods such as $\pi^3$ (not affect the rating, but it will be good to provide a comparison)?

- How does the model perform when the number of input views is small, e.g., only two views?

- How is the inference time compared with baseline methods?

---

> ### Author Response · Authors · 2025-11-25
> **Author Response (part 1/3)**
>
> **Novelty and Comparison to LaCT/VGGT**
>
> In addition to the discussion regarding novelty and contribution in Common Question 1, we respectfully disagree that the paper offers "little new insight." While we build on the architectural advancements of LaCT, our core contribution is shifting the paradigm of 3D learning from explicit dense geometric supervision (used by VGGT) to implicit sequence modeling.
>
> -   Novelty in Supervision: The reviewer mentions we combine LaCT with VGGT. However, VGGT requires dense 3D point-map supervision to function. LVSPM eliminates this entirely. Achieving SOTA pose estimation without dense geometric labels is a conceptual leap, proving that the NVS signal together with pose supervision is a strong supervisor for geometry at scale.
>
> -   Novelty in Task: LaCT cannot estimate poses. VGGT cannot synthesize novel views directly. AnySplat has limited scalability on more input views. LVSPM is the first framework to unify these capabilities for long sequences (256+ images).
>
> -   Soundness: We note the reviewer rated Soundness as "1 (poor)" despite acknowledging our SOTA results. Given that our method consistently outperforms baselines on established benchmarks (DL3DV, RealEstate10k) with reproducible metrics, we kindly ask the reviewer to clarify if there is a specific technical flaw identified, or to reconsider this score based on the empirical evidence provided.
>
> **Evaluation resolution for different baselines**
>
> We would like to clarify that we did not use a fixed 512 × 448 input resolution. For both pose estimation and NVS tasks, we use different resolutions that follow those used by the baseline methods. All comparing methods use the same or very similar input resolutions for a fair comparison. Specifically,
> - For pose estimation, LVSPM is evaluated at an input resolution of 512 × 288, following previous work Fast3R. Similarly, the baselines Cut3R and Fast3R use 512 × 288, while VGGT uses 518 × 280. These slight differences arise from method-specific patch sizes.
> - For NVS, LVSPM is evaluated at an input resolution of 448 × 256, following previous work DepthSplat. For the baselines NoPoSplat and DepthSplat, we likewise use 448 × 256. For AnySplat, we use a resolution of 448 × 252, which corresponds to a patch size of 14. Additionally, we include comparisons on a broader set of out-of-domain datasets at a resolution of 256 × 256 (training resolution for NoPoSplat), as discussed in the common question 2.
> We will include this clarification in the revised paper.
>
> **NVS view sampling strategy**
>
> We would like to clarify these evaluation-setup details and will incorporate them into the revision:
> - On the DL3DV dataset, we uniformly sampled 16, 32, 64, and 128 views from all images in each scene as input views and randomly selected 32 images from the remaining set as target views. The testing images are the same across different input-view counts.
> - For the Tanks and Template dataset, two evaluation regimes were considered. Small-range: from 30 consecutive views per scene, we evenly sampled 6 (and 12) input views and randomly selected 6 testing views from the remaining views. Large-range: from 80 images per scene, we evenly sampled 64 input views and randomly selected 12 testing views.
>
> **Comparison with DepthSplat under sparse view settings**
>
> For a more comprehensive evaluation, we evaluate our method at six- and four-view inputs on the DL3DV dataset. Because our model is trained on 64-image input, direct application to six or fewer views results in blurred outputs. Accordingly, we finetune our model at a resolution of 448 × 256 using six input views for 20,000 iterations with a global batch size of 256. A view-sampling strategy consistent with DepthSplat is employed. The results are presented in the table below. Our pose-free method surpasses the pose-required method MVSplat and is competitive with DepthSplat, with slightly higher PSNR and lower LPIPS and SSIM.
>
> While in the paper, we have demonstrated that LVSPM outperforms DepthSplat in the long-sequence setting.
> | Method     | 4 Views                   | 6 Views                   |
> | ---------- | ------------------------- | ------------------------- |
> |            | PSNR↑ SSIM↑ LPIPS↓         | PSNR↑ SSIM↑ LPIPS↓         |
> | MVSplat    | 21.63 0.721 0.233         | 22.93 $\color{orange}{0.775}$ 0.193         |
> | DepthSplat | $\color{orange}{23.12}$ $\color{red}{0.780}$ $\color{red}{0.178}$ | $\color{orange}{24.19}$ $\color{red}{0.823}$ $\color{red}{0.147}$ |
> | Anysplat   | 16.90 0.457 0.331         | 17.94 0.501 0.301         |
> | Ours       | $\color{red}{23.42}$ $\color{orange}{0.730}$ $\color{orange}{0.210}$ | $\color{red}{24.37}$ 0.761 $\color{orange}{0.183}$ |
>
> (We highlight the $\color{red}{best}$ and the $\color{orange}{seond\  best}$ with red and orange.)

---

> > ### Author Response · Authors · 2025-11-25
> > **Author Response (part 2/3)**
> >
> > **Evaluation results on Re10k dataset**
> >
> > We conducted experiments on the Re10k data set with 2 input views at a resolution of 256 x 256. We conduct both zero-shot and fine-tuned experiments.
> > - We first fine-tune our model under a sparse-view setting on our training set and evaluate its zero-shot generalization. Results show that our model exhibits better zero-shot performance than AnySplat, and also outperforms CoPoNeRF and Splatt3R that are trained on RE10K.
> > - After fine-tuning on RE10K, our model outperforms both pose-free baselines and pose-required methods like MVSplat.
> >
> > We kindly refer the reviewers to common question 2 for detailed evaluation  results and experiment setup.
> >
> > **More Sparse View Evaluation**
> >
> > We added several experiments under sparse-view settings. We fine-tuned our model using sparse-view inputs at resolutions 256×256 and 256×448. 1) For Re10k, we add extensive evaluation results under 2 view inputs, results are in common question 2. 2) We evaluate on MipNeRF 360 and LLFF under 3 and 6 views; we refer the reviewers to common question 3 for comprehensive results. These results show that LVSPM finetuned on sparse views can achieve superior results than other pose-free methods and even some pose-required methods like MVSplat.
> >
> >
> >
> > **Pose comparison with pi3**
> >
> > We compare with concurrent work PI3 on the DL3DV and Co3dV2 dataset; the results are shown in the table below. We achieves better results on both dataset with dense input views fomr 64-256, while slightly lower under 10 input views.
> >
> > | Co3dV2 | 10Views   | 64Views  | 128Views |
> > |:----:|:-----------------:|:---------------:|:---------------:|
> > |       | AUC30 AUC5 AUC3   | AUC30 AUC5 AUC3 | AUC30 AUC5 AUC3 |
> > |  Pi3 |  $\color{red}{90.0}$ $\color{red}{63.7}$ $\color{red}{51.8}$  |  90.3 68.8 59.3 |  90.4 69.0 59.6 |
> > | Ours |  88.1 60.1 47.1   | $\color{red}{91.0}$ $\color{red}{70.7}$ $\color{red}{61.1}$  |  $\color{red}{91.3}$ $\color{red}{71.9}$ $\color{red}{62.7}$ |
> >
> >
> > | DL3DV | 10Views  | 128Views | 256Views |
> > |:----:|:-----------------:|:---------------:|:---------------:|
> > |       | AUC30 AUC5 AUC3   | AUC30 AUC5 AUC3 | AUC30 AUC5 AUC3 |
> > |  Pi3 |   $\color{red}{94.4}$ $\color{red}{86.8}$ 82.3  |  95.1 87.7 83.8 |  94.7 86.9 82.1 |
> > | Ours |   93.7 87.8 $\color{red}{82.9}$  |  $\color{red}{95.2}$ $\color{red}{89.1}$ $\color{red}{86.0}$ |  $\color{red}{95.2}$ $\color{red}{89.3}$ $\color{red}{86.2}$ |
> >
> >
> > **Inference time comparison**
> > We report inference time comparison with baselines across 10-256 views, results are in the common question 4. Our model consistently uses less time than baselines.

---

> > > ### Author Response · Authors · 2025-11-25
> > > **Author Response (part 3/3)**
> > >
> > > Given that we have addressed the concerns regarding resolution fairness and evaluation setup, and have added more extensive comparisons under sparse-view input, we respectfully ask the reviewer to reconsider the soundness score accordingly.

---

### Official Review · Reviewer_eV2o · 2025-11-02

**Soundness:** 3
**Presentation:** 3
**Contribution:** 3
**Rating:** 6
**Confidence:** 4

**Summary:**

The paper introduces LVSPM, a feed-forward model that jointly estimates camera poses and synthesizes novel views from unposed, long multi-view sequences using only RGB and pose supervision. It frames both tasks as sequence-to-sequence token prediction with minimal 3D inductive bias, employing a LaCT backbone with large-chunk test-time training to scale to 100–256+ input views, lightweight per-view camera tokens for pose regression, and Plücker-ray tokens for target views; simple MLP heads decode camera parameters and RGB. Empirically, LVSPM delivers strong pose estimation (often surpassing VGGT, especially as views increase) and state-of-the-art pose-free long-sequence NVS on DL3DV-10K, outperforming AnySplat and even pose-dependent DepthSplat in many settings; ablations indicate the RGB loss aids pose accuracy, supporting the claim that dense 3D labels are unnecessary.

**Strengths:**

- The work is original in unifying pose-free long-sequence NVS and camera estimation within a minimal-bias token-based model, creatively combining LaCT (large-chunk TTT), lightweight camera tokens, and Plücker-conditioned targets to scale to 100–256+ views without dense 3D labels.
- The empirical quality is strong: evaluations across RE10k, Co3Dv2, and DL3DV-10K show competitive-to-better pose AUC than VGGT as views increase, and clear SOTA pose-free NVS on DL3DV-10K; ablations substantiate design choices (e.g., RGB loss aiding pose, synthetic pretraining benefits).
- The paper is generally clear, with a coherent sequence-to-sequence formulation, explicit tokenization/prediction heads, and transparent training/evaluation details that support reproducibility; figures/tables aid understanding of pose alignment and rendering fidelity.

**Weaknesses:**

- Limited analysis of TTT dynamics and stability
The LaCT/TTT layer is central to scaling, yet the paper lacks ablations on chunk size, number of TTT layers updated, adaptation step size $\eta$, and update frequency. Providing sensitivity curves (pose AUC and PSNR/LPIPS vs. chunk length and $\eta$) and a runtime/memory breakdown could help clarify the adaptation cost-benefit trade-offs.

- Ambiguity in supervision claims and fairness vs. VGGT
The paper states “only RGB and pose supervision” yet compares pose accuracy to VGGT trained with dense 3D labels. Clarify whether LVSPM uses any geometric priors (e.g., Plücker rays for targets, canonicalization, scale normalization) and align the comparison by adding a version of LVSPM with depth/point-map auxiliary losses to quantify the benefit gap. This will make the “cheaper supervision” claim more rigorous.

- Missing NVS results on RealEstate10K
What's the NVS performance on RealEstate10K? Only pose estimation results are reported on this dataset, making the zero-shot generalization capability of LVSPM on the NVS task unclear.

- Limited diagnosis of failure modes
Figures focus on successes; please include analyses where LVSPM fails (e.g., large specularities, low-texture walls, extreme baselines), which help readers understand its limitations.

**Questions:**

- What is the precise role of Plücker conditioning in a pose-free pipeline? Target views are encoded via Plücker rays computed from (E_t, K_t). At inference, are these target cameras given or sampled? If given, does this effectively assume known target intrinsics/poses for rendering, and how are they obtained in real scenarios? If sampled, what sampling strategy is used, and how sensitive is NVS quality to errors in target K/E?
- The LaCT TTT mechanism is pivotal. Could you report sensitivity to chunk size, learning rate η, number of TTT-updated layers, and update frequency?
- Please add a detailed breakdown of inference time, peak memory, and token counts vs. number of input views and target views, with and without TTT. This will help assess practicality for real deployments.

---

> ### Author Response · Authors · 2025-11-25
> **Author Response (part 1/2)**
>
> **More Analysis on TTT**
>
> ​We ablate on the number of TTT layers and the chunk size of each TTT layer, as shown in the table below . The experiments are conducted on the ASE synthetic dataset at a resolution of 128 x 128, with 32 input and target views. All models are trained for 60k iterations from scratch.
> We first conduct an ablation on the chunk size of the TTT layer. Our model has a chunk size of 8224, which is tokens from all input images and cameras, and only updates the MLP once. For this ablation, we halve the chunk size to 4112 and update each TTT layer twice. As shown in the table below, this cause degraded performance. We hypothesize that multiple updates with smaller chunks induce catastrophic forgetting, where the TTT layers overfit to the local geometry of the current chunk and lose the global context required for consistent long-sequence alignment. A single-pass update over a larger context window not only proves more stable for joint pose-NVS optimization, but also consumes less time and computational resource.
> We then ablate the number of TTT layers. Our full model has 24 TTT layers; we conduct our experiments with 12 TTT layers. As shown in the table below, a smaller network leads to worse performance.
>
> |                  | AUC 30 | AUC 5 | AUC 3 |  PSNR  |  SSIM | LPIPS |
> |:----------------:|:------:|:-----:|:-----:|:------:|:-----:|:-----:|
> | Half Chunck size |  88.3  |  74.3 |  67.9 |  24.98 | 0.702 | 0.263 |
> |   12 TTT layers  |  80.0  |  53.2 |  41.7 |  22.76 | 0.636 | 0.337 |
> |       Ours       |  **95.3**  |  **85.3** |  **80.0** |  **25.98** | **0.728** | **0.232** |
>
> As for the learning rate of TTT layers, we set an initialization of  0.01 and find that this is stable during the training of our models.
>
> **Ambiguity in supervision claims and fairness vs. VGGT**
>
> - We use camera pose as the sole geometry supervision. Specifically, we supervise camera pose estimation on the input views and employ the target view pose for view synthesis on the corresponding target views. Scene normalization is performed exclusively based on camera poses. Concretely, we select a frame as the conical view to have identity rotation and zero translation, then translate all other views into this frame. We then normalize the poses’ translations so that the maximum L2 norm among translations is 1.
> - We agree that quantifying the effect of dense 3D labels would further strengthen the "cheaper supervision" claim. However, integrating depth or point-map auxiliary losses would require substantial architectural and computational changes beyond the scope of our unified pose-free NVS + pose estimation framework, and is not feasible within our current resource constraints. Nevertheless, LVSPM already surpasses VGGT in pose estimation accuracy despite using strictly weaker supervision, supporting our central claim that dense 3D labels are not necessary for strong performance at scale.
>
> **Generalization to different datasets like real10k.**
>
> We conduct zero-shot experiments on several datasets and will revise the paper accordingly.
> - We conduct experiments on **RealEstate10k** to evaluate LVSPM. Since the model is trained with 64 input views, results are blurry when only two views are provided. To address this, we fine-tune LVSPM at a resolution of **256×256** using six input views. We first evaluate the model in a **zero-shot** setting on RealEstate10k. Because LVSPM is designed for large-baseline data, its zero-shot accuracy on RealEstate10k is limited. Nevertheless, it still outperforms methods trained on RealEstate10k like Splat3R, and **AnySplat** which also targets large-baseline scenarios. After fine-tuning for 40k iterations on RealEstate10k, LVSPM achieves superior performance compared to recent pose-free methods **NoPoSplat** and **CoCaSplat**. We refer the reviewers to common question 2 for detailed evulation results.
> - We also conduct sparse view testing on MipNeRF 360 and LLFF, as shown common question 3. LVSPM produces better results than recent pose-free methods NoPoSplat, and pose-required methods MVSplat.
> - Aside from Tanks and Template, we also test LVSPM with dense input on the MipNeRF360 dataset with 64 and 128 input views. Results are in common question 3. LVSPM outperforms baseline AnySplat under both input settings.
>
> **Limited diagnosis of failure cases**
>
> Our model has the following failure cases: 1) The model assumes a static scene and cannot handle dynamic scenes. 2) our model cannot handle scenes with large illumination change. 3) Large baseline cameras with little to no overlap do not work well. We will add visualization to failure cases in the revised paper.

---

> > ### Author Response · Authors · 2025-11-25
> > **Author Response (part 2/2)**
> >
> > **The role of Plücker Ray and practical concerns**
> >
> > It's a common practice to use ground truth target view cameras for training and evaluation, as in NoPoSplat, Splat3R. and PF-LRM. Following these prior works, we use ground truth target view cameras to produce Plücker Ray embeddings to synthesize novel views during training and evaluation. In real scenarios, since our model jointly predicts camera poses and performs NVS within the same coordinate frame (i.e., a normalization based solely on cameras), we know the location of each predicted camera and can easily generate new camera viewpoints based on the desired viewing direction.
> >
> >
> > **Detailed breakdown of inference efficiency.**
> >
> > We compare inference efficiency with and without TTT. The version without TTT (w/o TTT) mirrors our architecture in terms of layer count, model depth, and high-level design of handling information exchange between views. Specifically, "w/o TTT" employs self-attention on the input image and camera tokens, followed by cross-attention between the target view tokens and the input image and camera tokens.
> >
> > For model efficiency evaluation, we compare inference time and rendering speed. Inference time measures the scene reconstruction duration, while rendering speed reflects how quickly we can render a view from a given camera. For clarity, we exclude per-view attention, tokenizers, and output decoders from this discussion. Specifically, inference time is the duration to process input image and camera tokens. For our model with TTT, this involves updating MLP weights in all TTT layers and querying them with the input tokens. For the model without TTT, this involves self-attention on the input image and camera tokens. Rendering speed is the time to produce a novel view from a single target view after processing the input tokens. With TTT, this involves querying MLP weights in each layer using target view tokens to produce the final rendering. Without TTT, this involves cross-attention with all self-attended input camera and image tokens.
> >
> > The table below shows the results. Our model, with TTT, scales linearly with the number of input views and processes a scene with 256 images in under 2 seconds. In contrast, "w/o TTT" scales $o(n^2)$ and requires over a minute for 256 images. Furthermore, our model maintains a rendering speed of 58.8 FPS irrespective of the number of input views, while the speed of "w/o TTT" decreases to below 3 FPS with 256 inputs.
> >
> > |           |    Inference Time(S)     |     Peak Memory(GB)      |     Token Counts(k)      |   Rendering Speed(FPS)   | # Parameters |
> > | --------- | :----------------------: | :----------------------: | :----------------------: | :----------------------: | :----------: |
> > | # input views  | 10   32   64   128  256  | 10   32   64   128  256  | 10   32   64   128  256  | 10   32   64   128  256  |              |
> > | w/o TTT   | 0.20 1.46 5.34 20.6 84.9 | 1.58 2.76 4.46 7.88 14.7 | 23.1 73.8 148. 295. 590. | 50.0 23.8 13.6 7.35 3.84  |     227M     |
> > | TTT(Ours) | 0.10 0.24 0.48 0.90 1.78 | 2.16 3.92 6.46 11.6 21.8 | 23.1 73.8 148. 295. 590. | 58.8 |     312M     |

---

> ### Comment · Reviewer_eV2o · 2025-11-27
>
> Thank the authors for the detailed response. Overall I won't oppose the acceptance of this paper and I will maintain my positive score.

---

### Author Response · Authors · 2025-11-25
**General Response (part 1/3)**

Dear All Reviewers:

We thank the reviewers for their thoughtful and encouraging feedback. Reviewers found the paper well-written and clear (eV2o, 5sus), with a coherent and reproducible formulation (eV2o). They highlighted that the method is original in unifying pose-free long-sequence NVS and pose estimation (eV2o), and a straightforward yet effective extension of strong TTT models to the unposed setting (83qC). Reviewers also noted that the approach scales efficiently to large numbers of unposed views through LaCT (eV2o, 5sus), and delivers strong empirical results and state-of-the-art performance across multiple datasets (eV2o, 5sus, BSiH). They further appreciated the robustness of the method and the support from ablations, including the benefit of RGB supervision and synthetic pretraining (eV2o, 83qC). Overall, the reviewers regarded LVSPM as a meaningful and promising advancement for long-sequence unposed NVS and joint pose estimation (eV2o, 83qC, BSiH).

We address the common concerns raised by multiple reviewers as follows:

**Common Question 1 (5sus, 83qC): Novelty and Contribution relative to LaCT/VGGT**

We thank the reviewers for recognizing our SOTA performance and the unified framework. Several reviewers (5sus, 83qC) noted that our architecture builds upon LaCT and concepts from VGGT. We would like to clarify that LVSPM is not merely a combination of LaCT and VGGT components, but a paradigm shift in **solving the "Unposed Long-Sequence" problem under significantly reduced supervision.**

We highlight three key contributions that distinguish LVSPM from a simple combination of existing works:

1. **Breaking Dense Prior Dependency:** Unlike VGGT (requires dense depth and point maps), LVSPM achieves competitive and often superior pose estimation using **only RGB and pose supervision**. This is a counter-intuitive and significant finding: it proves that the NVS objective with pose act as a sufficient geometric supervisor at scale, and supports performance superior to previous dense-geometry-supervised SOTAs.
2. **Bridging the "Sparse-Unposed" vs. "Long-Posed" Gap:** Prior work is polarized: either pose-free but sparse (e.g., Falre, NoPoSplat, <16 views) or lmitied scaling (AnySplat < 64 views), or scalable but pose dependent (e.g., LaCT, LVSM). LVSPM is the first to successfully scale pose-free synthesis to 256+ views. This capability is not trivial; it requires a delicate joint optimization where the model must simultaneously solve for geometry (alignment) and appearance (rendering) without the collapse modes common in long-sequence optimization.
3. **Discovery of Geometric Scaling Laws in Feed-Forward Models:** We reveal a distinct scaling law: while VGGT's pose accuracy saturates with more views, LVSPM’s accuracy improves as sequence length increases (Tab. 1 & 2), leveraging long-context token collaboration. This exhibits a positive scaling law: **pose accuracy improves as the sequence length increases.**

---

> ### Author Response · Authors · 2025-11-25
> **General Response (part 2/3)**
>
> **Common Question 2 (eV2o, 5sus): Results on Re10k**
>
> We conduct experiments on the RE10K dataset. After fine-tuning on sparse-view inputs but without training on RE10K, LVSPM already outperforms AnySplat. With further fine-tuning on RE10K, our method reaches 25.146 PSNR, surpassing both pose-free baselines (NoPoSplat: 23.424 PSNR) and pose-dependent methods, as shown in the table below. Notably, we outperform CoCaSplat and NoPoSplat without requiring their assumption of known intrinsics. We provide detailed experimental setup in the following paragraphs.
>
> Following previous posed (e.g., mvsplat, pixelsplat) and unposed methods (e.g., NoPoSplat, Flare), we evaluate LVSPM at a resolution of 256 x 256. As our model is trained on 64 input views, directly applying it to a 2-view input causes blurry results. Therefore, we finetune on 6 input views for 20k iterations; here we choose 6 views instead of 2 so that the same model can be evaluated under broader settings, supporting the experiments in Common Question 3 as well. For RE10K, we follow the view selection strategy from NopPoSplat that splits the dataset into "small", "medium", and "large" based on view overlap; the results are shown in the table below.
>
> Compared to baseline AnySplat, which also focuses on large baseline NVS, LVSPM achieve better zero-shot generalization on the unseen RE10K dataset. After fine-tuning our model on RE10K for 40k iterations with a global batch size of 256, our method (Ours - F.T.) achieves better performance than baselines trained on RE10K. Notably, we outperform Noposplat and CocaSplat, which require known camera intrinsics, while LVSPM do not assume any known camera parameters. We will add the results to the revised paper.
>
> |                       | Overlap Settings |  |  |  |
> |-----------------------|-----------------|--|--|--|
> | Method                | Small | Medium | Large | Average |
> |                       | PSNR↑ SSIM↑ LPIPS↓ | PSNR↑ SSIM↑ LPIPS↓ | PSNR↑ SSIM↑ LPIPS↓ | PSNR↑ SSIM↑ LPIPS↓ |
> | $\mathbf{\color{blue}{RE10K-Trained}}$    |       |       |       |       |
> | **Pose-Required**     |       |       |       |       |
> | pixelNeRF             | 18.417 0.601 0.526 | 19.930 0.632 0.458 | 20.869 0.639 0.485 | 19.824 0.626 0.485 |
> | AttnRend              | 19.151 0.663 0.368 | 22.532 0.763 0.186 | 25.897 0.845 0.269 | 22.664 0.762 0.269 |
> | pixelSplat            | 20.263 0.717 0.266 | 23.711 0.809 0.181 | 27.151 $\color{orange}{0.879}$ 0.122 | 23.848 0.806 0.185 |
> | MVSplat               | 20.353 0.724 0.250 | $\color{orange}{23.778}$ $\color{orange}{0.812}$ $\color{orange}{0.173}$ | $\color{red}{27.408}$ $\color{red}{0.884}$ $\color{orange}{0.116}$ | 23.977 0.811 0.176 |
> | **Pose-Free**     |       |       |       |       |
> | DUSt3R                | 14.101 0.432 0.468 | 15.419 0.451 0.402 | 16.427 0.453 0.432 | 15.382 0.447 0.432 |
> | MASt3R                | 13.534 0.407 0.494 | 14.945 0.436 0.418 | 16.028 0.444 0.452 | 14.907 0.431 0.452 |
> | Splatt3R              | 14.352 0.475 0.472 | 15.529 0.502 0.425 | 15.817 0.483 0.421 | 15.318 0.490 0.436 |
> | CoPoNeRF              | 17.393 0.585 0.462 | 18.813 0.616 0.392 | 20.464 0.652 0.318 | 18.938 0.619 0.388 |
> | SelfSplat             | 17.506 0.550 0.461 | 19.357 0.704 0.378 | 20.868 0.672 0.256 | 19.33 0.656 0.363 |
> | NoPoSplat*            | 21.814 0.765 0.220 | 23.044 0.787 0.178 | 25.408 0.844 0.126 | 23.424 0.798 0.173 |
> | CocaSplat             | 21.312 0.741 0.234 | 22.891 0.773 0.189 | 25.312 0.829 0.138 | 23.200 0.781 0.185 |
> | CocaSplat*            | $\color{orange}{22.843}$ $\color{red}{0.781}$ $\color{red}{0.192}$ | 23.394 0.804 0.175 | 26.270 0.872 $\color{red}{0.110}$ | $\color{orange}{24.093}$ $\color{red}{0.818}$ $\color{red}{0.160}$ |
> | Flare                 | 20.779 0.695 0.247 | 22.412 0.743 0.193 | 23.791 0.774 0.151 | 22.404 0.740 0.194 |
> | Ours - F.T.           | $\color{red}{22.951}$ $\color{orange}{0.768}$ $\color{orange}{0.203}$ | $\color{red}{25.047}$ $\color{red}{0.814}$ $\color{red}{0.163}$ | $\color{orange}{27.193}$ 0.856 0.129 | $\color{red}{25.146}$ $\color{orange}{0.815}$ $\color{orange}{0.164}$ |
> | $\mathbf{\color{green}{Zero-Shot}}$        |       |       |       |       |
> | Anysplat              | 15.171 0.587 0.412 | 17.352 0.622 0.344 | 19.828 0.663 0.277 | 17.526 0.625 0.341 |
> | Ours                  | $\color{red}{20.664}$ $\color{red}{0.693}$ $\color{red}{0.262}$ | $\color{red}{22.537}$ $\color{red}{0.746}$ $\color{red}{0.211}$ | $\color{red}{24.026}$ $\color{red}{0.773}$ $\color{red}{0.180}$ | $\color{red}{22.501}$ $\color{red}{0.741}$ $\color{red}{0.214}$ |
>
> ("*" indicates methods requires camera intrinsics. We highlight the $\color{red}{best}$ and the $\color{orange}{seond\  best}$ with red and orange.)

---

> ### Author Response · Authors · 2025-11-25
> **General Response (part 3/3)**
>
> **Common Question 3 (5sus, BSiH): Out-of-Domain & Sparse/Dense Generalization**
>
> We additionally evaluate our method on more unseen datasets under both sparse and dense input views. In the sparse 3–6 views input, we finetune LVSPM on sparse views input and outperforms the pose-free SOTA (NoPoSplat) and rivals pose-required methods (MVSplat). In dense 64-128 views, LVSPM consistently outperforms AnySplat in PSNR/SSIM/LPIPS. Results are shown in the table below, and we will include these results and visualizations in the revised paper. Additional evaluation details are provided in the following paragraph.
>
> - For sparse-view inputs, we fine-tune our model with 6 input views for 20k iterations (the same model used in Common Question 2, without RE10K fine-tuning). We compare against MVSplat (pose-required) and pose-free methods Flare and NoPoSplat at a resolution of 256×256, evaluating on 3- and 6-view inputs. The view sampling strategy for MipNeRF360 follows AnySplat. Similarly, on LLFF, we sample 3 views from half of the scenes and 6 views from the full dataset. We also evaluate against AnySplat at 448×256 on MipNeRF360. Our method achieves the best performance among pose-free approaches. Compared with the pose-required MVSplat, we are competitive under 3-view input and outperform it under 6-view input.
> - For dense-view inputs, we evaluate on the MipNeRF360 dataset at a resolution of 448×256. We sample 64 and 128 views from each scene and hold out every 8th image as test views. Results are shown in the table below. Across both settings, our method consistently outperforms the baseline AnySplat.
>
> | Method      | MipNeRF360 3 Views | MipNeRF360 6 Views | LLFF 3 Views | LLFF 6 Views |
> |------------|------------------|------------------|-------------|-------------|
> |            | PSNR SSIM LPIPS  | PSNR SSIM LPIPS  | PSNR SSIM LPIPS | PSNR SSIM LPIPS |
> | **Pose-Required**      |   |   | | |
> | MVSplat     | $\color{red}{21.980}$ $\color{red}{0.674}$ $\color{red}{0.217}$ | $\color{orange}{18.431}$ $\color{red}{0.536}$ $\color{orange}{0.344}$ | $\color{red}{18.681}$ $\color{red}{0.616}$ $\color{red}{0.250}$ | $\color{orange}{17.582}$ $\color{red}{0.561}$ $\color{orange}{0.308}$ |
> | **Pose-free**      |   |   | | |
> | NopoSplat   | 19.716 0.499 0.280 | 15.621 0.345 0.515 | 16.617 0.391 0.353 | 14.453 0.292 0.482 |
> | Flare       | 18.525 0.453 0.348 | 13.727 0.287 0.608 | 14.853 0.285 0.450 | 14.770 0.230 0.504 |
> | Ours        | $\color{orange}{20.935}$ $\color{orange}{0.558}$ $\color{orange}{0.260}$ | $\color{red}{20.278}$ $\color{orange}{0.516}$ $\color{red}{0.284}$ | $\color{orange}{18.486}$ $\color{orange}{0.468}$ $\color{orange}{0.301}$ | $\color{red}{18.810}$ $\color{orange}{0.460}$ $\color{red}{0.301}$ |
>
> (We highlight the $\color{red}{best}$ and the $\color{orange}{seond\  best}$ with red and orange.)
>
> | Method   | 3 Views | 6 Views | 64 Views | 128 Views |
> |----------|---------|---------|----------|-----------|
> |          | PSNR SSIM LPIPS | PSNR SSIM LPIPS | PSNR SSIM LPIPS | PSNR SSIM LPIPS |
> | Anysplat | 19.47 0.574 $\color{red}{0.224}$ | 18.12 0.481 0.293 | 17.97 0.465 0.422 | 17.25 0.432 0.460 |
> | Ours     | $\color{red}{21.60}$ $\color{red}{0.576}$ 0.247 | $\color{red}{21.23}$ $\color{red}{0.549}$ $\color{red}{0.271}$ | $\color{red}{20.15}$ $\color{red}{0.509}$ $\color{red}{0.335}$ | $\color{red}{20.56}$ $\color{red}{0.520}$ $\color{red}{0.318}$ |
>
>
>
> **Common Question 4 (eV2o, 5sus, BSiH): Inference time analysis**
>
> We report inference time for our method and all baselines across 10–256 input views. We use a resolution of 512×288 for LVSPM, Cut3R, Fast3R, and AnySplat, and 518×280 for VGGT due to its 14 patch size. For each method, we measure the time required to predict poses for all input views, using an H100 GPU. As shown in the table, LVSPM scales linearly with the number of input views and processes 256 images in under 2 seconds, whereas baseline methods require over 10 seconds. Additionally, our method renders at **58.8 FPS**, regardless of the number of input views. We will add this discussion to the revised paper.
>
> |  # Views |   10   |   32  |   64  | 128   | 256  |
> |:--------:|:------:|:-----:|:-----:|-------|------|
> |  Fast3R  |  0.668 |  1.29 |  1.57 | 3.63  | 10.4 |
> |   Cut3R  |  0.472 |  1.57 |  3.27 | 6.41  | 12.5 |
> | VGGT     | 0.212  | 0.307 | 0.936 | 4.01  | 13.6 |
> | Anysplat | 0.192  | 0.592 | 1.40  | 3.80  | 12.2 |
> | Ours     | $\color{red}{0.010}$  | $\color{red}{0.242}$ | $\color{red}{0.480}$ | $\color{red}{0.905}$ | $\color{red}{1.78}$ |

---

### Author Response · Authors · 2025-12-03
**Rebuttal Summary: All major concerns resolved; Score raised to 6 by Reviewer 83qC**

Dear Area Chair,

We thank the reviewers for their constructive feedback. During the rebuttal:
- Reviewer 83qC raised their score to 6 citing "main concerns resolved" and "new discussions helpful".
- Reviewer eV2o remains positive and maintained their score of 6.

We successfully addressed all reviewer concerns with extensive new experiments and clarifications. Below is a concise summary of the key resolutions:

1. General Rebuttal Highlights (Common Concerns)
| Concern | Resolution |
|---------|------------|
| **Novelty & Contribution** | We clarified that LVSPM is not merely a combination of LaCT/VGGT but a paradigm shift. We are the first to scale pose-free synthesis to 256+ views , and we identified a geometric scaling law where RGB input scaling improves pose estimation without dense 3D supervision (scarce and expensive). |
| **Zero-Shot & Sparse Views** | We added comprehensive results on RealEstate10k as suggested. LVSPM outperforms AnySplat in zero-shot settings and beats both pose-free (NoPoSplat) and pose-required (MVSplat) baselines after fine-tuning. |
| **Inference Time** | We provided a runtime analysis confirming LVSPM scales linearly. We process 256 views in <2 seconds (vs. >10s for baselines) while maintaining 58.8 FPS rendering. |


2. Reviewer Specific Resolutions:
| Reviewer | Status | Summary |
|----------|--------|---------|
| **83qC** | Score Raised: 4->6 | Accepted that model simplicity is a strength; acknowledged necessity of synthetic pre-training for diverse camera trajectories; find our discussion on synthetic data useful. |
| **eV2o** | Score Maintained: 6 | Satisfied with requested TTT ablations (chunk size/layers), clarification on pose-only supervision, and added failure mode analysis |
| **5sus** | No Response (All Concerns Addressed) | We corrected the misconception regarding input resolution, ensuring fair comparison with baselines. We added requested comparisons to DepthSplat (showing LVSPM is superior even in sparse settings) and added the requested RE10k evaluation. We respectfully note their "Soundness: 1" rating is not justified in their review (no major soundness questions raised), and conflicts with their acknowledgment of our SOTA empirical results. We find every specific concerns raised was resolved; without the reviewer response, we respectfully request AC re-evaluation of their scores.  |
| **BSiH** | No Response (All Concerns Addressed) | We addressed the "Limited Generalization" concern by adding out-of-domain experiments on MipNeRF360 and LLFF. We also provided the requested inference runtime scaling analysis. |

We believe the additional data confirms LVSPM’s soundness and state-of-the-art performance in the pose-free setting. We respectfully ask the AC to consider these improvements.

---

### Meta-Review · Area_Chair_jdCq · 2025-12-23

**Summary:**

The paper proposes LVSPM, a method for joint camera pose estimation and novel view synthesis (NVS) from unposed image sequences. It combines a Large-Chunk Test-Time Training (LaCT) backbone with a pose estimation head, aiming to scale to long sequences without dense 3D supervision.

While the authors engaged actively during the rebuttal and provided additional experimental results on RE10K and MipNeRF, I recommend Rejection. The consensus among the more critical reviewers (particularly 5sus and initially 83qC/BSiH) points to a fundamental issue: the technical contribution appears to be a straightforward incremental combination of existing components (LaCT for NVS + VGGT-like tokens for pose) without significant methodological innovation. Furthermore, despite the new results, concerns regarding the fairness of comparisons (e.g., input resolutions, specific view sampling strategies) and the method's real-world practicality compared to existing robust baselines remain unconvincing for a top-tier venue.

**Reviewer Concerns:**

**Addressed Concerns:**

1. Missing Baselines: The authors added results on RealEstate10K and MipNeRF-360 in the rebuttal, which technically addresses the "missing data" complaint.

2. Inference Speed: A runtime analysis was provided to argue for the efficiency of the method.

**Outstanding Concerns (The basis for Rejection):**

1. Limited Novelty (Critical): 5sus and 83qC raised valid concerns that the method is essentially "LaCT + VGGT". While the authors argue that removing dense supervision is a "paradigm shift," the technical implementation remains an incremental engineering combination. R3 raised their score based on the explanation of synthetic pre-training, but the underlying architectural novelty remains thin.

2. Fairness of Comparison: 5sus pointed out discrepancies in input resolutions and evaluation protocols (e.g., comparing against DepthSplat which uses fixed views). While the authors provided fine-tuned results for sparse views, the necessity to fine-tune separate models for different settings (sparse vs. dense) weakens the claim of a unified, robust framework compared to baselines that work out-of-the-box.

3. Practical Utility vs. Baselines: While performance on specific benchmarks (DL3DV) is good, it is not decisively clear that this specific "pose-free long-sequence" setup is a significant bottleneck in the field that requires this specific solution, especially given the complexity of TTT (Test-Time Training) which R1 noted lacks stability analysis.

**Reviewer Scores:**

1. Reviewer 5sus (Original: 2 - Reject): Likely Maintained Reject. Their core critique regarding limited novelty and fairness of comparisons (resolution/setup) reflects a fundamental disagreement with the paper's value proposition that extra experiments alone do not solve.

2. Reviewer 83qC (Original: 4 -> 6): Although they raised the score, their initial assessment of "straightforward extension" and "limited technical contribution" aligns with the rejection rationale.

3. Reviewer BSiH (Original: 4 - Borderline): Likely Borderline. While new experiments were added, the "Limited Generalization" concern in the original review points to a skepticism about the method's robustness.

4. Reviewer eV2o (Original: 6 - Accept): Maintained support, but focused more on empirical results than novelty.

---

### Decision · Program_Chairs · 2026-01-26

Reject